# CartoonNet: Cartoon Parsing with Semantic Consistency and Structure Correlation

Jian-Jun Qiao
Southwest Jiaotong University
Chengdu, China
qjjai56@gmail.com

Meng-Yu Duan
Southwest Jiaotong University
Chengdu, China
duanmengyu369@gmail.com

Xiao Wu*
Southwest Jiaotong University
Chengdu, China
wuxiaohk@gmail.com

Yu-Pei Song
Southwest Jiaotong University
Chengdu, China
yupei-song@my.swjtu.edu.cn

## Abstract

Cartoon parsing is an important task for cartoon-centric applications, which segments the body parts of cartoon images. Due to the complex appearances, abstract drawing styles, and irregular structures of cartoon characters, cartoon parsing remains a challenging task. In this paper, a novel approach, named CartoonNet, is proposed for cartoon parsing, in which semantic consistency and structure correlation are integrated to address the visual diversity and structural complexity for cartoon parsing. A memory-based semantic consistency module is designed to learn the diverse appearances exhibited by cartoon characters. The memory bank stores features of diverse samples and retrieves the samples related to new samples for consistency, which aims to improve the semantic reasoning capability of the network. A self-attention mechanism is employed to conduct consistency learning among diverse body parts belong to the retrieved samples and new samples. To capture the intricate structural information of cartoon images, a structure correlation module is proposed. Leveraging graph attention networks and a main body-aware mechanism, the proposed approach enables structural correlation, allowing it to parse cartoon images with complex structures. Experiments conducted on cartoon parsing and human parsing datasets demonstrate the effectiveness of the proposed method, which outperforms the state-of-the-art approaches for cartoon parsing and achieves competitive performance on human parsing.

## CCS Concepts

• **Computing methodologies** → **Image segmentation**.

## Keywords

Cartoon Parsing, Memory Bank, Graph Attention Network

*Corresponding author: Xiao Wu

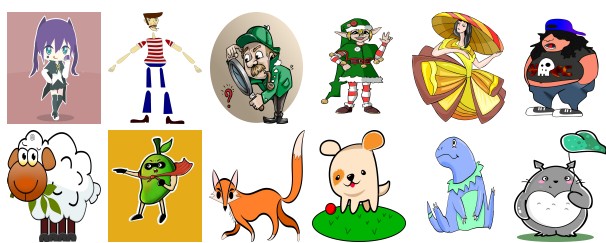

**Figure 1: Cartoon parsing remains a challenging task due to the various styles, complex appearances and abstract structures of cartoon images. The diversity and complexity of cartoon images pose great difficulties for cartoon parsing.**

**ACM Reference Format:**
Jian-Jun Qiao, Meng-Yu Duan, Xiao Wu, and Yu-Pei Song. 2024. CartoonNet: Cartoon Parsing with Semantic Consistency and Structure Correlation. In *Proceedings of the 32nd ACM International Conference on Multimedia (MM '24), October 28–November 1, 2024, Melbourne, VIC, Australia.* ACM, New York, NY, USA, 9 pages. https://doi.org/10.1145/3664647.3680879

## 1 Introduction

Cartoon characters are an important part of culture and media with their diverse appearances and abstract structures. From human-like cartoon characters to animal-like cartoon images, cartoons exhibit a wide range of styles. The task of cartoon parsing, which aims to segment different parts of cartoon characters, holds substantial importance for various applications such as cartoon animation, content creation and virtual worlds. Due to the inherent variations in cartoon appearances and lack of uniformity in cartoon structures, identifying diverse visual appearances and complex structures of cartoons is a challenging task.

Cartoon images have various visual appearances and abstract spatial structures, which are illustrated in Fig. 1. Unlike real-world humans, the styles of cartoon characters may vary depending on author or cultural background, which results in the diversity and complexity of cartoon images. For human-like cartoon characters, their appearances and structures are quite different. The properties of cartoon images make the body parts such as limbs, head and body of human-like cartoon characters abstract and complicated, which are not consistent with real-world humans. Animal-like cartoon images are more complex due to diverse animal categories. For different animal species such as fish, mammals, and reptiles, their visual appearances and body structures are different when

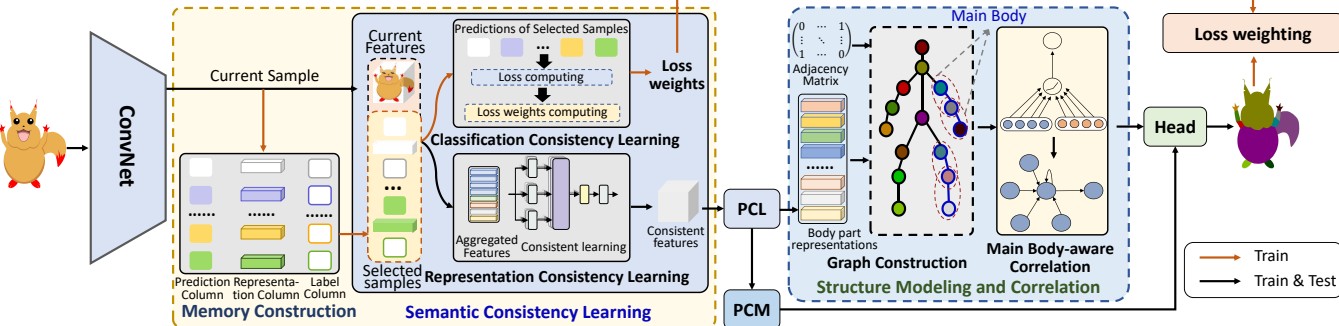

**Figure 2: The framework of the proposed CartoonNet. It mainly consists of memory-based semantic consistency and main body-aware structure correlation. The memory-based semantic consistency utilizes a memory bank to achieve consistency learning across different samples. The main body-aware structure correlation focuses on correlation of body structures, modeling the structure of cartoons. PCL and PCM refer to pixel correlation learning and part correlation modeling [23], which are employed to solve the irregularities of cartoon images.**

presented in the form of cartoon images. Therefore, cartoon parsing remains a challenging task.

For image parsing, existing methods have primarily focused on human parsing and achieved significant progress. Previous human parsing methods have explored hierarchical structure learning [12, 34, 35, 43], pose estimation-guided learning [19, 41] and the integration of self-correction strategy [14]. However, due to the significant differences between real humans and cartoon images, these human parsing methods exhibit limited performance in cartoon parsing. With the development of cartoon related applications, the advanced deep learning methods [23, 32] have been applied to cartoon parsing. They employ multi-scale learning structures, and leverage pixel and part correlation to infer irregular cartoon structures. However, the diversity and complexity of cartoon images remain a challenge.

In this paper, CartoonNet is proposed to address the diversity and complexity of cartoon images in cartoon parsing. It augments cartoon parsing by designing a memory-based semantic consistency structure and a graph attention network-based structure correlation module. The framework is illustrated in Fig. 2. The semantic consistency module encodes and learns the diverse appearances displayed by cartoon characters. It designs a memory-based learning structure to conduct consistency learning of diverse body parts. The memory-based semantic consistency learns the visual differences of cartoon characters, which aims to alleviate the rich diversity of cartoon images and improve generalization ability. The structure correlation approach focuses on modeling and learning the intricate and irregular structures exhibited by cartoon characters. It leverages graph attention networks (GATs) [2, 11, 31] and a main body-aware mechanism to facilitate structural learning and correlation. By modeling the structural features of cartoon images across the main bodies, the method focuses on important parts of cartoon images to alleviate the structural complexity of cartoon parsing. Experiments conducted on cartoon parsing and human parsing datasets demonstrate the effectiveness of the proposed approach, which outperforms the state-of-the-art methods for cartoon parsing and achieves competitive results on human parsing. The contributions are summarized as follows:

- CartoonNet is proposed to address the challenges of visual diversity and structure complexity in cartoon images, which designs

a memory-based semantic consistency module and a main body-aware structure correlation strategy for cartoon parsing.
- A memory-based learning structure is proposed to facilitate appearance learning, which captures the inconsistent appearances of different cartoon characters and utilizes self-attention mechanism to achieve semantic consistency among diverse body parts.
- A main body-aware structure correlation method is proposed to model the structural information of cartoon characters with complex structures, which employs the graph attention networks and a main body-aware mechanism to focus on the important structures.
- The proposed method achieves state-of-the-art performance on the cartoon parsing datasets, which demonstrates the effectiveness of CartoonNet.

## 2 Related Work

### 2.1 Human Parsing

Human parsing [5, 16, 29], the task of segmenting human body parts in images, has achieved significant progress. Previous human parsing methods develop approaches such as hierarchical structure learning [12, 34, 35, 43], pose estimation-guided learning [19, 41], and self-correction strategy [14], achieving satisfactory results on recognition of body parts of humans. However, applying human parsing methods to cartoon parsing presents challenges due to fundamental differences. Cartoons are known for their complexity, diverse styles, irregular lines, unique color schemes, and abstract representations, varying greatly based on creators and cultural contexts. Human parsing models [15, 18] are designed on real-world human images, making them unsuitable for cartoon parsing, where characters often have unconventional body proportions and articulations. Cartoon parsing necessitates dedicated research and specialized algorithms to address its unique characteristics, emphasizing accuracy and effectiveness in dealing with the diversity and complexity of cartoon images.

### 2.2 Cartoon Parsing

The task of cartoon parsing has gained increasing attention in recent years due to its relevance in various multimedia applications. Researchers have proposed several approaches to address the challenges posed by diverse and complex cartoon appearances

and structures. Early works in this domain primarily focused on rule-based methods and handcrafted feature extraction techniques. These approaches, while capable of capturing some aspects of cartoon images, often struggle with the diversity and complexity of cartoon imagery. The emergence of deep learning techniques, particularly the convolutional neural networks (CNNs) [1, 17, 25, 27], has significantly advanced the field of cartoon parsing. DFPNet [32] applies feature pyramid network to cartoon dog parsing, which alleviates the varied scales in cartoon parsing. However, it ignores the correlations of visual cues and structural features in cartoon characters. CPNet [23] recognizes the irregular cartoon structures and body parts with different semantics but have visually akin appearances, by introducing a pixel and part correlation learning structure. However, it ignores the semantic consistency of diverse cartoon images and the structure correlation of complex cartoon characters. Although CNN-based methods demonstrate improved performance in recognizing cartoon characters, addressing the diversity and complexity of cartoon images remains a challenge.

## 2.3 Memory-Augmented Networks

Memory-augmented neural networks [36, 37] have shown promise in capturing and utilizing long-term dependencies in data. These networks [13, 24] employ an external memory structure that can be read from and written to, enabling the storage and retrieval of information over extended sequences. In the context of cartoon parsing, memory-augmented networks can offer a promising potential for encoding and learning the diverse appearances exhibited by cartoon characters. Despite the growing interest in memory-augmented networks, their potential for cartoon parsing remains unexplored. The ability to store appearance information and perform semantic consistency learning opens possibilities for addressing individual visual differences. Therefore, this paper proposes a novel approach applicable to cartoon parsing to address the issue of significant sample diversity in cartoon parsing through a memory-based structure.

## 3 Cartoon Parsing

### 3.1 Framework

CartoonNet is proposed to address the problems of cartoon parsing, which are caused by visual diversities and structural complexities of cartoon characters. A two-fold strategy integrating appearance learning and structural modeling is introduced to capture features of diverse visual appearances and model complex structural representations. The framework is illustrated in Fig. 2, in which ConvNet [10] is employed to encode and extract features for cartoon images. The memory bank is adopted to store features of diverse samples with a three-column structure. The stored features are matched to current images by measuring feature similarity, which aims to parse current samples by recalling previous experiences. The previous experiences are integrated into current samples by adopting self-attention mechanism, which achieves semantic consistency among diverse samples. To learn the correlation of complex body parts, a main body-aware structure correlation strategy is designed. It incorporates graph attention networks and a main body-aware mechanism to facilitate part correlation. The graph attention networks model the structural representations and capture important relations among the body parts. The main body-aware mechanism

learn important body parts that reflect the complex structural properties of the cartoon images.

## 3.2 Memory Bank Construction

Previous cartoon parsing methods only consider multi-scale learning, as well as pixel and part correlation among a single cartoon character [23, 32]. They have difficulties in identifying diverse cartoon images. Therefore, the first significant component of the approach is to encode and learn the visual appearances of diverse cartoon characters. A memory-based learning structure is proposed to address the problems caused by visual diversity, which aims to achieve semantic consistency among different semantic parts. The memory-based consistency learning includes two parts, representation consistency and classification consistency. The representation consistency is designed to conduct consistency learning on semantic parts from different cartoon characters at high-dimensional semantic feature level. The classification consistency is proposed to achieve consistency learning of semantic parts from various cartoon characters at the semantic recognition level, which leverages the constraint information.

Given a cartoon image $X$, $F \in \mathbb{R}^{C \times H \times W}$ is the encoded image features outputted by ConvNet. $C$ denotes the number of channels. $H$ and $W$ refer to height and width of the feature maps, respectively. A memory bank with three columns is defined, where the prediction column stores the two-dimensional prediction maps, the representation column saves one-dimensional feature representations, and the label column one retains labels. Specifically, $F$ is fed into a decoder with three convolution layers [4] to obtain the two-dimensional feature maps $F_{mem} \in \mathbb{R}^{C_n \times H \times W}$, where $C_n$ denotes the number of classes. $F_{mem}$ is then stored in the prediction column of the memory bank. Additionally, the operation of information aggregation with global adaptive max pooling is performed on each feature map of $F$, which outputs a feature vector $F_v \in \mathbb{R}^{C \times 1 \times 1}$. $F_v$ is the representation of $X$ and it is stored in the representation column. Label $L$ corresponding to $X$ is stored in the label column of the memory bank, which aims to provide supervision for the semantic consistency learning. In the initial stage, all three columns of the memory structure are initialized as zeros. During the training process, image features of each iteration are stored in the memory bank. When the memory is full, the earliest stored features are deleted from memory to ensure that the memory bank has space for features of new iterations, which aims to continuously update the memory bank throughout the training process.

During each iteration, $F$ not only gets stored in the memory structure but also undergoes further forward operations. A memory selection operation is performed to choose samples from the memory structure. These selected samples have some similarities to current sample and they are used to achieve semantic consistency, which aims to enhance the semantic reasoning of the network for targets sharing similar characteristics. Specifically, when inputting a new sample, the network selects similar samples from the memory structure. By leveraging the knowledge acquired from these learned samples, the network accurately infers and identifies current sample. To achieve memory selection, during each training iteration, a similarity calculation is performed between current $F_{mem}^c$ and the stored $F_{mem}^m$ within the memory structure. This is done using the

Jian-Jun Qiao, Meng-Yu Duan, Xiao Wu, and Yu-Pei Song

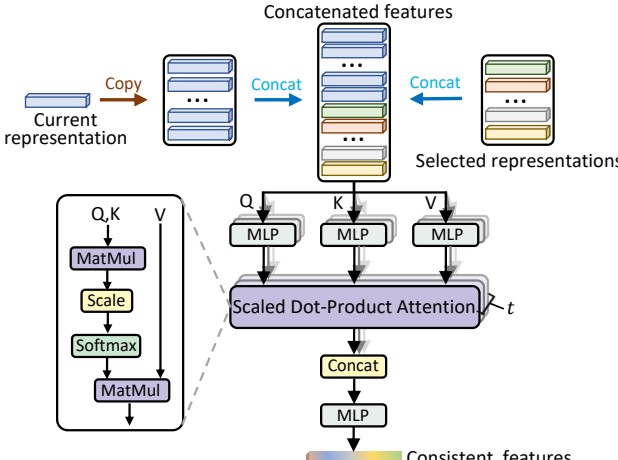

**Figure 3: Memory-based representation consistency learning.**

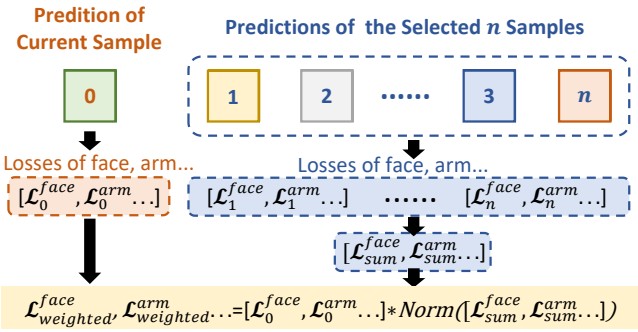

**Figure 4: Memory-based classification consistency learning.**

Euclidean distance, which retrieves $n$ samples from the memory structure.

## 3.3 Representation Consistency Learning

The problem of inconsistent appearances among different cartoon images poses great challenges for cartoon parsing. To achieve consistency learning for the body parts that belong to the same class but from different samples, the network initially performs consistency learning at the representation level. The framework of the representation consistency learning is illustrated in Fig. 3, which involves the integration and consistency of feature representations of current sample and feature representations of the relevant samples retrieved from the memory bank. Multi-head self-attention [30] is employed to achieve consistency learning among the semantic parts from different samples. It ensures that current sample and the samples stored in the memory structure are associated and consistent at the feature representation level.

To conduct the representation consistency learning, $F$ is mapped into 1-dimensional feature representation $F_v^c$ with a global adaptive max pooling operation. Through the memory selection, $n$ feature representations $F_v^{m,n}$ that represents the retrieved samples are selected for current sample. $F_v^c$ is then duplicated $n$ times and concatenated with $F_v^{m,n}$ to aggregate features from diverse samples. The above operations are formulated as follows:

$$F_v^{cat} = Cat(Copy(Gap(F)), Selection(M)) \quad (1)$$

where $F_v^{cat}$ is the concatenated features. $Gap(\cdot)$ denotes global adaptive max pooling operation. $Copy(\cdot)$ is the duplication operation. $Cat(\cdot)$ refers to concatenation operation. $Selection(\cdot)$ denotes the memory selection operation that retrieves the related samples from memory $M$. Subsequently, multi-perceptual (MLP) layers are adopted to create queries, keys and values for multi-head self-attention. It is formulated as,

$$Q = F_v^{cat}W^Q, \quad K = F_v^{cat}W^K, \quad V = F_v^{cat}W^V \quad (2)$$

where $W^Q$, $W^K$, and $W^V$ are learnable weight matrices that project the input vectors $F_v^{cat}$ into queries $Q$, keys $K$, and values $V$, respectively. Multi-head self-attention mechanism calculates the output vectors $F_v^{out}$ as a weighted sum of the input vectors $F_v^{cat}$, where the weights are determined by the attention scores. The attention

scores are calculated as,

$$Attention(Q, K, V) = softmax\left(\frac{QK^T}{\sqrt{d_k}}\right)V \quad (3)$$

where dot products of the query $Q$ with all keys $K$ are computed and scaled by the scaling factor $\frac{1}{\sqrt{d_k}}$. $d_k$ is the dimension of query. The softmax operation generates the attention scores. The dot products of the attention score and value $V$ associate the representations of different samples.

Multi-head attention mechanism has $t$ heads and it performs the attention calculation in parallel for each head. The outputs of the heads are concatenated and projected back to the original dimension of $F_v^{cat}$, which is formulated as,

$$head_i = Attention(QW_i^Q, KW_i^K, VW_i^V) \quad (4)$$

$$F_v^{out} = MultiHead(Q, K, V) = Cat(head_1, \ldots, head_t)W^O \quad (5)$$

where $W_i^Q$, $W_i^K$, $W_i^V$, and $W^O$ are parameter matrices.

The representation consistency learning with self-attention mechanism allows the model to attend to different representations of the same input sequence when calculating the output vectors $F_v^{out}$. The consistent features, $F_v^{out}$, is fused to the original features $F$ as,

$$F_{rc} = Cat(F, F \odot Sig(F_v^{out})) \quad (6)$$

where $Sig(\cdot)$ denotes sigmoid operation that aims to map the values of vectors into calibration weights. $\odot$ refers to element-wise multiplication.

## 3.4 Classification Consistency Learning

To achieve semantic consistency, supervisions are required for diverse body parts during the training process. Therefore, in addition to representation consistency learning, classification consistency learning is proposed and performed through loss weighting, which achieves semantic consistency among current sample and the samples retrieved from the memory bank. The labels of current sample and the retrieved samples are utilized to provide supervision.

With the memory selection operation, predictions $F_{mem}^{m,n}$ of the retrieved $n$ samples are obtained, which are related to features $F$ of current sample $X$. $F$ is mapped into category dimension for semantic prediction. The decoder of DeepLabV3+ [4] is adopted for the mapping, which is widely used in previous human parsing methods. It is formulated as,

$$F_{mem}^c = Decoder(F, F') \quad (7)$$

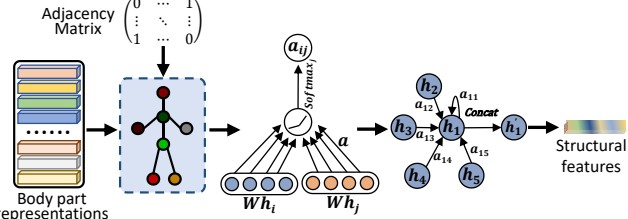

**Figure 5: GAT-based structure correlation.**

where $F'$ denotes the shallow features from ConvNet that provide spatial details for the deep features $F$.

To ensure that the network consistently recognizes the body parts that belong to the same class in $F_{mem}^c$ and $F_{mem}^{m,n}$, the classification consistency is performed by calibrating the recognition results of $F_{mem}^c$ using the recognition results of $F_{mem}^{m,n}$. As can be seen in Fig. 4, for each sample in $F_{mem}^{m,n}$, losses of each sample are calculated for each category such as face and arm, which outputs $n$ loss values for the same category. The loss values of the same category are summed for the corresponding category, aggregating the losses from different samples. Subsequently, all summed losses are normalized and mapped to weights. The weights are multiplied by the loss of each category in $F_{mem}^c$. Consequently, the recognition of each category in $F_{mem}^c$ is calibrated by the corresponding category in $F_{mem}^{m,n}$, which achieves classification consistency learning. It is formulated as follows:

$$\mathcal{L}_{cls} = CE_{cls}(F_{mem}^{m,n}, L^n) \tag{8}$$

$$\mathcal{L}_{wts} = Norm(Sum_{cls}(\mathcal{L}_{cls})) \tag{9}$$

$$\mathcal{L}_{weighted} = CE_{cls}(F_{mem}^c, L) \odot \mathcal{L}_{wts} \tag{10}$$

where $CE_{cls}(\cdot)$ means modified cross-entropy loss function, which is used for loss computation of each class. $Sum_{cls}(\cdot)$ denotes gathering the losses of the same class. $Norm(\cdot)$ refers to normalization on the summed losses of different categories. $F_{mem}^c$ and $F_{mem}^{m,n}$ are predictions of the current sample and the retrieved samples, respectively. $L$ and $L^n$ are their corresponding labels, respectively.

## 3.5 Structure Correlation

In addition to appearance learning, structural learning is also crucial for cartoon parsing. Previous methods consider the correlation of different parts, but they overlook the learning of strongly related local structures. The pipeline of the main body-aware mechanism is illustrated in Fig. 6, in which some local structures exhibit high correlations, such as the face, hair, and hat. Similarly, the thighs, shins, and shoes, or the palms, upper arms, and forearms, are also strongly related. Therefore, this paper introduces the main body-aware recognition strategy, which focuses on structural modeling of strongly correlated structures to explore complex structures of cartoon images.

Given $F_{rc}$, pixel correlation learning (PCL) is adopted to capture pixel correlations of the cartoon image [23]. Feature aggregation is then performed using global average pooling along the channel dimension to obtain a one-dimensional feature representation. The one-dimensional feature representation is divided into $n_c$ feature representations, which are treated as nodes of graph $\mathcal{G} = (\mathcal{V}, \mathcal{E})$. $n_c$ represents the number of categories. $\mathcal{V}$ denotes the set of nodes that correspond to body parts of cartoon characters. $\mathcal{E}$ refers to

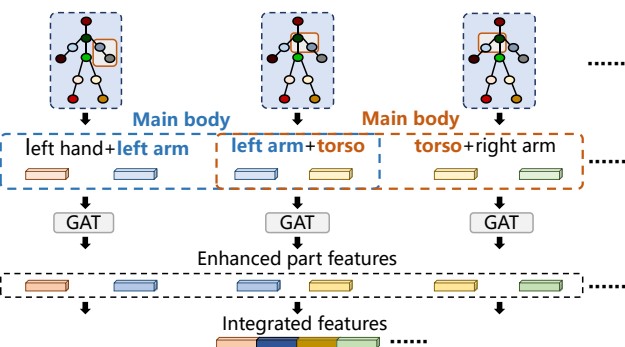

**Figure 6: Main body-aware mechanism.**

the set of edges that indicates the connection relations of adjacent body parts. For the construction of edge $\mathcal{E}$, the adjacency matrix that indicates the connections of semantic parts is computed to associate body parts.

GAT layers [31] are constructed to model the graph $\mathcal{G}$. The framework of the GAT-based structure correlation is illustrated in Fig. 5. The network first computes the attention score $a_{ij}$ between nodes $i$ and $j$. For a node $i$ in graph $\mathcal{G}$, the attention of its neighbor $j$ is defined as,

$$a_{ij} = LeakyReLU(a^T Cat(Wh_i, Wh_j)) \tag{11}$$

where $a$ is a learnable vector. $h_i$ and $h_j$ are the feature vectors of nodes $i$ and $j$. $LeakyReLU(\cdot)$ is the activation function used for non-linearity. $Cat(\cdot)$ allows the model to consider the features of both node $i$ and $j$. The features are transformed with weight matrix $W$, which learns the importance of different node pairs.

Softmax function is applied to the attention scores, which normalizes them to obtain weights $\alpha_{ij}$. The attention coefficient $\alpha_{ij}$ represents the importance of node $j$ when updating the representation of node $i$, which is calculated as,

$$\alpha_{ij} = \frac{\exp(a_{ij})}{\sum_{k \in \mathcal{N}_i} \exp(a_{ik})} \tag{12}$$

where $\mathcal{N}_i$ is the set of neighbors of node $i$.

The new representation of node $i$, $h_i'$, is calculated as a weighted sum of the transformed features of its neighbors, with weights given by the attention coefficients $\alpha_{ij}$. This allows each node to selectively attend to its neighbors, capturing the importance of different connections in the graph. It is formulated as,

$$h_i' = \sigma\left(\sum_{j \in \mathcal{N}_i} \alpha_{ij} Wh_j\right) \tag{13}$$

where $\sigma$ denotes to a nonlinearity operation [31].

The ultimately updated node representation $h_i'$ is obtained by employing multi-head attention mechanism [30]. $S$ independent attention mechanisms perform the transformation described in Equation 13. The output features are concatenated as the ultimately updated node representation $h_i'$.

Unlike the typical GAT, the GATs built are hierarchical, which are designed to model structural representations of global and local regions. Firstly, a GAT layer is adopted to model the pre-constructed graph structure, updates all node data and returns the results. Subsequently, the subgraph structures are built by merging strongly related nodes among the $n_c$ nodes, and GAT is then applied to model the strongly related nodes and return the results. As can be

seen in Fig. 6, the graph of the left hand, left arm and torso is treated as a main body graph because they are strongly connected through the left arm. By feeding the paired nodes associated with the left arm into GAT for learning, the structural information related to the left arm is emphasized. The network facilitates the recognition of specific regions by focusing on the main body structure. After that, the learning results of various main bodies are aggregated and the integrated results are combined with the initially returned results from the global graph, producing the ultimate output. The output of the structure correlation is fused with the input features, resulting in stronger structural features that are used for accurate structural learning, which is formulated as,

$$h'_i = SC_{GATs}(Divide(Gap(PCL(F_{rc})))) \tag{14}$$

$$F_{rc-main} = Head(F_{rc} \oplus h'_i, PCM(PCL(F_{rc}))) \tag{15}$$

where $SC_{GATs}(\cdot)$ denotes the structure correlation function based on GATs. $Divide(\cdot)$ is the operation dividing the one-dimensional feature representation into $n_c$ nodes. $Head(\cdot)$ means a stacked module of concatenation and a convolution layer. $PCM(\cdot)$ denotes part correlation modeling [23].

## 3.6 Loss Functions

The objective function is composed of the proposed classification consistency loss, pixel correlation loss, part correlation loss, spatial loss and edge loss, which is formulated as,

$$\mathcal{L}_{obj} = \mathcal{L}_{cc} + CE(F^c_{mem}, L) + CE(F_{pcl}, L) + \\ CE(F_{rc-main}, L) + \mathcal{L}_{spatial} + \mathcal{L}_{edge} \tag{16}$$

where $CE(\cdot)$ is the cross-entropy loss function. $CE(F^c_{mem}, L)$ is adopted to predict segmentation results using features from ConvNet. $CE(F_{pcl}, L)$ and $CE(F_{rc-main}, L)$ are used to learn pixel and part correlations [23]. $F_{pcl}$ is obtained by using a convolution layer to map the output of PCL into predictions. $\mathcal{L}_{spatial}$ and $\mathcal{L}_{edge}$ are utilized to learn spatial information [20, 23, 28] and edge information [14, 26]. $\mathcal{L}_{cc}$ is the proposed classification consistency loss, which is defined as follows:

$$\mathcal{L}_{cc} = (CE_{cls}(F^c_{mem}, L) + CE_{cls}(F_{pcl}, L) + \\ CE_{cls}(F_{rc-main}, L)) \odot \mathcal{L}_{wts} \tag{17}$$

## 4 Experiments

## 4.1 Datasets and Evaluation Metrics

**CartoonSet.** The CartoonSet dataset [23] comprises 2,229 cartoon images, which are divided into 1,530 images for training, 510 for testing, and 189 for validation. It includes diverse samples such as drawings of children, illustrations with brief strokes and animated characters. The cartoon images are annotated and the body parts of cartoon characters are categorized into 24 classes.

**Cartoon dog.** The Cartoon dog dataset [32] contains 965 images of cartoon dogs. It consists of 773 images for training and 192 images for testing. This dataset focuses on cartoon dog parsing, with eight classes such as head and body for labeling.

**LIP.** LIP [8] is a human parsing dataset consisting of 50,462 single-person images. It is divided into 30,462 training images, 10,000 testing images, and an additional 10,000 images for validation. It provides annotations for 19 classes, which include categories such as face and right arm.

**Evaluation Metrics.** Following previous approaches [15, 22, 23, 32], the utilized evaluation metric is Mean Intersection over Union (Mean IoU). Mean IoU computes the average intersection-to-union ratio between the predicted body parts and ground truth. Additionally, Mean Accuracy (Mean Acc.) is employed to calculate per-class accuracy and Pixel Accuracy (Pixel Acc.) is used to compute the accuracy of the correctly predicted pixels.

## 4.2 Implementation Details

To implement the proposed approach, Stochastic Gradient Descent (SGD) is employed as the optimizer with a momentum of 0.9 and a weight decay of 5e-4. Following previous methods [3, 7, 21, 38, 39, 42], the "poly" learning rate strategy is utilized, which is defined as $lr = lr_{base} \times (1 - \frac{C_n}{T_n})^{power}$. $lr$ and $lr_{base}$ are the current learning rate and the base learning rate, respectively. $power = 0.9$. $C_n$ and $T_n$ represent the current iteration number and the total iteration number, respectively. For data augmentation, random left-right flipping with a 0.5 probability and random scaling strategy are applied. For a fair comparison on CartoonSet, CartoonNet is trained with a batch size of 8 and image size of 384×384. The number of training epochs is 150 and the learning rate for CartoonSet is set to 7e-3. For Cartoon dog dataset, CartoonNet uses a learning rate of 1e-3, and the training images are resized to 384×384. All methods adopt single-scale evaluation on CartoonSet and Cartoon dog dataset. For LIP dataset, CartoonNet is trained with a learning rate of 7e-3, with training images resized to 473×473. The number of training epochs is 150. The multi-scale evaluation approach [14, 15] is employed for LIP dataset.

## 4.3 Comparison with the State-of-the-Art Approaches

To evaluate the performance of the proposed approach, it is compared with the state-of-the-art human parsing approaches and cartoon parsing methods. DeepLabV3+ [4], a model not specialized for cartoon parsing, has limited performance in the domain of cartoon parsing. DFPNet [32] is designed for cartoon parsing and it boosts the performance by introducing a multi-scale learning mechanism. But it primarily focuses on scale variations, resulting in limited performance. Methods like HHP [35] and CNIF [34] employs GNN to model the structures of cartoon image, which improves the results with structural modeling. However, their hierarchical structure learning is designed for real humans with consistent and regular structures. When segmenting the irregular and complex body structures of cartoon characters, the performance is limited [23]. CE2P [26] leverages global context information and edge details to improve the results. But it fails to account for the visual diversity and structural complexity of cartoon images, resulting in limited performance. CDGNet [15] uses class distributions as spatial constraints, outperforming most of the methods. But it ignores the structural modeling and faces challenges when segmenting cartoon images with complex structures. SCHP [14] achieves improved results using a self-correction strategy, which helps differentiate visually complex body parts. Nonetheless, it still faces challenges caused by the diversity and complexity of cartoon characters. CPNet [23] significantly improves the performance of cartoon parsing by addressing the problems caused by the irregular structures with

**Table 1: Comparison on CartoonSet.**

| Method | Pixel Acc. | Mean Acc. | Mean IoU |
|---|---|---|---|
| DeepLabV3+ [4] | 87.28 | 63.42 | 50.12 |
| DFPNet [32] | 88.21 | 65.33 | 51.71 |
| HHP [35] | 87.51 | 64.42 | 51.98 |
| CE2P [26] | 88.06 | 66.55 | 52.90 |
| CNIF [34] | 87.74 | 66.01 | 53.21 |
| CDGNet [15] | 88.11 | 67.98 | 53.99 |
| SCHP [14] | 88.63 | 69.36 | 55.44 |
| CPNet [23] | 89.42 | 69.90 | 57.02 |
| CartoonNet (Ours) | **89.61** | **70.59** | **57.84** |

**Table 2: Comparison on Cartoon dog dataset.**

| Method | Pixel Acc. | Mean Acc. | Mean IoU |
|---|---|---|---|
| Mask R-CNN [9] | 89.21 | 57.78 | 50.56 |
| CE2P [26] | 92.63 | 76.43 | 65.32 |
| DFPNet [32] | 93.50 | 79.40 | 68.39 |
| SCHP [14] | 94.05 | 81.15 | 71.22 |
| CDGNet [15] | 94.09 | 80.03 | 71.44 |
| CPNet [23] | 94.32 | 82.60 | 72.28 |
| CartoonNet (Ours) | **94.64** | **84.01** | **74.05** |

its pixel and part correlation learning module. However, CPNet [23] mainly focuses on independent cartoon characters, which makes it less effective for segmenting diverse cartoon characters. Compared to the previous methods, CartoonNet achieves the best results on CartoonSet. The memory-based semantic consistency learning module recalls previous experiences to improve the generalization ability. The main body-aware mechanism focuses on the important local regions to improve structural correlation learning. CartoonNet makes a deep exploration of semantic consistency and structure correlation across diverse and complex cartoon images, addressing the challenges posed by cartoon characters of multiple styles.

To further evaluate the performance of the proposed method, it is tested on Cartoon dog dataset. Overall, the performance of previous human parsing methods is limited when tested on Cartoon dog dataset. The limitations are caused by the differences between abstract cartoon images and real-world humans. Cartoon parsing methods like CPNet [23] perform better because it accounts for the properties of cartoon images. But the diversity and complexity of cartoon images remain challenges, which limits the performance. Compared to previous methods, CartoonNet achieves the state-of-the-art performance on Cartoon dog dataset, as listed in Table 2. The specific design of CartoonNet effectively alleviates the challenges posed by the diverse visual appearances and abstract structures of cartoon dogs. The results listed in Table 2 further demonstrate the effectiveness of the proposed method.

To verify the generalization ability of CartoonNet, it is tested on human parsing dataset LIP. The comparative results are listed in Table 3. Methods like DeepLabV3+ [4], OCR (ResNet101) [40], OCR (HRNetV2-W48) [40] and HRNetV2 [33] are common semantic segmentation methods. Although these methods achieve outstanding performance on segmentation datasets such as Cityscapes [6], their performance on LIP is limited due to the lack of specific designs for human parsing. Compared to the common semantic segmentation approaches, the human parsing methods like HHP [35], SCHP [14] and CDGNet [15] achieve better performance. Their components such as GNN-based hierarchical structure learning module [34, 35], self-correction strategy [14] and class distribution learning

**Table 3: Comparison on LIP dataset.**

| Method | Pixel Acc. | Mean Acc. | Mean IoU |
|---|---|---|---|
| DeepLabV3+ [4] | n/a | n/a | 52.09 |
| CE2P [26] | 87.37 | 63.20 | 53.10 |
| CorrPM [44] | 87.68 | 67.21 | 55.33 |
| OCR (ResNet101) [40] | n/a | n/a | 55.60 |
| HRNetV2 [33] | n/a | n/a | 55.90 |
| OCR (HRNetV2-W48) [40] | n/a | n/a | 56.65 |
| CPNet [23] | 88.29 | 68.41 | 57.21 |
| CNIF [34] | 88.03 | 68.80 | 57.74 |
| HHP [35] | **89.05** | 70.58 | 59.25 |
| SCHP [14] | n/a | n/a | 59.36 |
| CDGNet [15] | 88.86 | **71.49** | **60.30** |
| CartoonNet (Ours) | 88.59 | 69.57 | 58.27 |

mechanism [15] are designed for real-world humans. Although the proposed CartoonNet is designed for cartoon parsing, it achieves competitive performance on LIP dataset. Compared to common semantic segmentation methods, the performance of CartoonNet on LIP dataset is unexpected, which outperforms methods like OCR (HRNetV2-W48) [40] and HRNetV2 [33] by a large margin. CartoonNet also outperforms the cartoon parsing method CPNet [23] significantly on LIP dataset, which proves its generalization ability. The performance of CartoonNet on LIP dataset demonstrates that the proposed semantic consistency and structure correlation approach is effective for human parsing as well.

## 4.4 Ablation Studies

To validate the effect of the proposed method, ablation studies of semantic consistency and structure correlation are conducted. The results are listed in Table 4. Compared to the baseline model, the incorporation of representation consistency and classification consistency leads to a significant improvement in Mean IoU metric. The memory-based consistency learning strategy facilitates the semantic consistency of the body parts that belong to the same class but from diverse cartoon images. When adopting the main body-aware structure correlation module, the performance is further boosted. The structure correlation module focuses on important local regions with main body-aware mechanism and captures important relations among body parts with graph attention network. The integration of the memory-based consistency module and the main body-aware structure correlation achieves the best results, confirming that the proposed method is effective in addressing challenges posed by diverse appearances and complex structures of cartoon parsing. To further verify the effect of the proposed modules, ablation studies on Cartoon dog dataset are conducted and the results are listed in table 5. Compared to the baseline model, CartoonNet with the proposed modules significantly boosts the performance, which proves the effectiveness of the memory-based semantic consistency strategy and main body-aware structure correlation module.

To verify the effect of the memory bank, ablation studies of the size of the memory bank and the number of selected similar samples from the memory bank are conducted. In Table 6, CartoonNet + M400 represents training CartoonNet with a memory storage size of 400 samples. 1530 is the number of images of the training set in CartoonSet dataset. It can be seen from the table that the storage size of the memory bank has little impact on the accuracy. CartoonNet + N3 represents selecting three similar samples in each iteration,

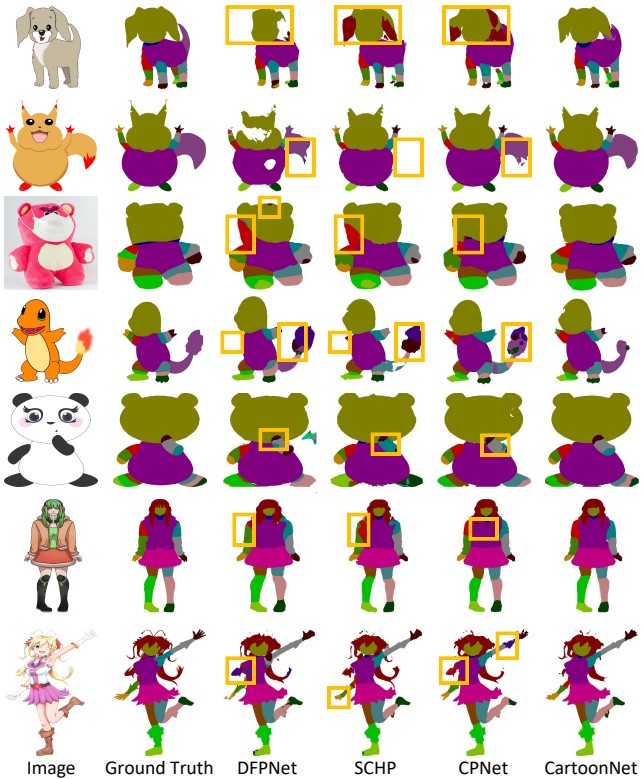

Image    Ground Truth    DFPNet    SCHP    CPNet    CartoonNet

**Figure 7: Visualization comparisons on CartoonSet.**

**Table 4: Effect of the proposed modules on CartoonSet dataset.**

| Method | Pixel Acc. | Mean Acc. | Mean IoU |
|---|---|---|---|
| Baseline | 89.19 | 68.91 | 56.12 |
| Baseline + RC | 89.33 | 70.24 | 57.11 |
| Baseline + CC | 89.51 | 69.71 | 57.08 |
| Baseline + RC + CC | 89.49 | 70.50 | 57.47 |
| Baseline + Main | 89.48 | 69.61 | 57.26 |
| Baseline + RC + CC + Main | **89.61** | **70.59** | **57.84** |

**Table 5: Effect of the proposed modules on Cartoon dog dataset.**

| Method | Pixel Acc. | Mean Acc. | Mean IoU |
|---|---|---|---|
| Baseline | 94.09 | 82.22 | 72.13 |
| Baseline + RC | 94.47 | 83.05 | 73.24 |
| Baseline + CC | 94.37 | 81.91 | 73.09 |
| Baseline + RC + CC | 94.50 | 83.19 | 73.69 |
| Baseline + Main | 94.44 | 82.69 | 72.77 |
| Baseline + RC + CC + Main | **94.64** | **84.01** | **74.05** |

and the data in the table indicates that the best result is obtained when the number of the selected samples is 5.

## 4.5 Qualitative Evaluation

To qualitatively evaluate the proposed method, the visualization results of CartoonNet and the state-of-the-art approaches are illustrated in Fig. 7. For diverse cartoon characters, existing methods tend to make recognition errors when segmenting body parts that belong to the same class but exhibit significant visual and structural differences. For example, the heads of panda and dog are quite different. Previous methods like CPNet [23] have difficulties in segmenting diverse cartoon images as they primarily focus on

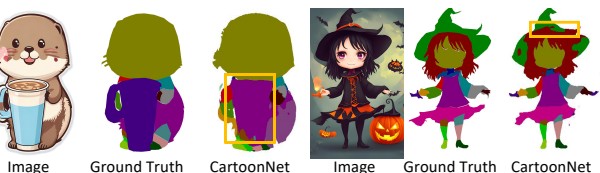

Image    Ground Truth    CartoonNet        Image    Ground Truth    CartoonNet

**Figure 8: Examples of failure cases.**

**Table 6: Ablation studies of the memory size and the number of selected samples.**

| Method | Pixel Acc. | Mean Acc. | Mean IoU |
|---|---|---|---|
| CartoonNet + M400 | **89.61** | 70.59 | **57.84** |
| CartoonNet + M800 | 89.51 | 70.48 | 57.45 |
| CartoonNet + M1200 | 89.56 | **70.94** | 57.31 |
| CartoonNet + M1530 | 89.41 | 70.45 | 57.67 |
| CartoonNet + N3 | 89.49 | 70.04 | 57.15 |
| CartoonNet + N5 | **89.61** | 70.59 | **57.84** |
| CartoonNet + N7 | 89.67 | **70.85** | 57.47 |
| CartoonNet + N9 | 89.51 | 69.64 | 57.40 |

visual appearances within a single cartoon character. The proposed method conducts semantic consistency learning across different cartoon characters with a memory bank to recall experiences for previous samples, achieving better results. Previous methods also have limited performance when the targets have intricate and irregular structures such as the hair of the girls, as they lack specialized recognition structures for the irregularities in cartoon characters. Among the previous methods, CPNet [23] performs better by considering the irregularities of cartoon characters. However, it still falls short in segmenting complex structures as it ignores the main bodies. The proposed method introduces the main body-aware strategy, which addresses the challenges related to complex structures by focusing on core local regions and capturing important relations among body parts.

Although CartoonNet achieves the state-of-the-art performance, it still faces some challenges. The failure examples are illustrated in Fig. 8. When the bodies of cartoon characters are significantly occluded by other objects such as cups, it is difficult for CartoonNet to correctly segment the cartoon image. In addition, if the adjacent body parts have similar appearances, for example, the black hat and black hair, CartoonNet may make some error predictions.

## 5 Conclusion

In this paper, a cartoon parsing method named CartoonNet is proposed to address the challenges posed by diverse appearances and complex structures of cartoon characters. It consists of a semantic consistency approach based on a memory bank and a main body-aware structure correlation module based on graph attention networks. The memory-based consistency learning achieves semantic consistency among diverse cartoon characters, which explores deep relations of the body parts that have different appearances but belong to the same class. The main body-aware structure correlation boosts the performance of complex structures by focusing on the core body parts and capturing important relations among the body parts. By integrating the semantic consistency strategy and the structure correlation module, the proposed method achieves the state-of-the-art results in cartoon parsing.

## Acknowledgments

This work was supported in part by the National Natural Science Foundation of China (Grant No. 62372387), and Key R&D Program of Guangxi Zhuang Autonomous Region, China (Grant No. AB22080038, AB22080039).

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
