# OpenReview forum: "CartoonNet: Cartoon Parsing with Semantic Consistency and Structure Correlation"
_acmmm.org/ACMMM/2024/Conference — MM2024 Poster_

### Official Review · Reviewer_JpYR · 2024-05-05

**Rating:** 5
**Confidence:** 3

**Summary:**

The paper "CartoonNet: Cartoon Parsing with Semantic Consistency and Structure Correlation" introduces a novel approach, CartoonNet, for parsing cartoon characters. It focuses on recognizing and segmenting various parts of cartoon characters by integrating semantic consistency and structure correlation to address the challenges posed by the visual diversity and structural complexity of cartoon images.

**Strengths:**

Innovative Approach: CartoonNet proposes a unique method that combines semantic consistency and structure correlation to effectively parse cartoon characters with complex appearances and irregular structures.
State-of-the-Art Performance: The proposed method achieves state-of-the-art performance on cartoon parsing datasets, demonstrating its effectiveness in handling the multifaceted appearances of cartoon imagery.
Main Body-Aware Structure Correlation: The incorporation of a main body-aware structure correlation method enables the recognition of cartoon characters with substantial structural complexity, addressing the inherent challenges in cartoon structures.

**Limitations:**

Domain Specificity: The approach is tailored specifically for cartoon parsing and may not generalize well to other types of image parsing tasks.
Dataset Dependency: The performance of CartoonNet is evaluated on specific cartoon parsing datasets, and its generalizability to a broader range of cartoon styles and structures may need further exploration.

**Suitability:**

3

---

### Official Review · Reviewer_7YHu · 2024-05-24

**Rating:** 3
**Confidence:** 3

**Summary:**

This paper proposes a new network, CartoonNet, for recognizing and segmenting different parts of cartoon characters. Two modules are proposed to address the challenges of diverse semantic representations and complex human structures in cartoon characters. And good results were achieved on both the cartoon parsing dataset and the human parsing dataset.

**Strengths:**

1. The memory based semantic consistency mechanism is very convincing, and can fully solve problems with diverse semantic expressions both theoretically and experimentally.
2. By learning the internal structure of the target, it is possible to more accurately encode the correlation between the target's local features and the overall features.
3. In addition, CartoonNet has achieved good results on both cartoon parsing datasets and single person human parsing datasets.

**Limitations:**

Although I strongly support the author's idea of using memory to address external diversity within the same category, I have significant opinions on its implementation methods and paper descriptions.
1. from the perspective of theoretical novelty, memory based semantic feature matching or unified technology is usually applied in Re-ID, Track, VOS and other fields, and in the field of part analysis, VLParts[1] also adopted a similar sample matching approach in the open-world part segmentation challenges. And semantic feature consistency and body structure analysis are not uncommon in the field of Human Parsing, such as UniParser[2] and sematree[3]. Therefore, I believe that the paper has novelty in its ideas, but it is not outstanding.

[1] Sun, Peize, et al. "Going denser with open-vocabulary part segmentation." Proceedings of the IEEE/CVF International Conference on Computer Vision. 2023.
[2] Chu, Jiaming, et al. "UniParser: Multi-Human Parsing with Unified Correlation Representation Learning." arXiv preprint arXiv:2310.08984 (2023).
[3] Ji, Ruyi, et al. "Learning semantic neural tree for human parsing." Computer Vision–ECCV 2020: 16th European Conference, Glasgow, UK, August 23–28, 2020, Proceedings, Part XIII 16. Springer International Publishing, 2020.

2.My biggest negative feedback on this paper comes from its writing and expression, as the description of the connections between modules in the paper is too poor, which makes it difficult for me to grasp the author's original intention. Even if MM accepts this article, I suggest making significant revisions. I will not provide the detailed description of writing problem, like formula formatting misalignment and inappropriate size and position of icon text, but they need to solve. The description of each module in the paper is too independent, and the description of the model process in Figure 2 is too rough, making it difficult for readers to clearly understand the training and inference process of the model.
In addition, I also have doubts about the design scheme of each module. For example, in the representation consistency module, it is necessary to unify the semantics of the current feature with the semantics of the reference feature. So, how can we obtain the semantic results of the current image feature without a semantic segmenter in front of the model? If using groundtruth to achieve consistency, does it indicate that the consistency module is not working during inference? These should be described in detail in Chapter 3 and Figure 2. The model structure design is also relatively simple and common, lacking novelty.

**Suitability:**

3

---

### Official Review · Reviewer_yK2i · 2024-05-27

**Rating:** 2
**Confidence:** 3

**Summary:**

This paper studies the cartoon paring problem, and proposes the CartooNet to recognize and segment various parts of cartoon characters, where semantic consistency and structure correlation are integrated to address the visual diversity and structure complexity.
A memory-based semantic consistency module is designed to center on encoding and learning the multifaceted appearances exhibited by
cartoon characters.
A structure correlation module is proposed to enable structural learning and correlating, allowing it to recognize and parse cartoon images with significant complexity.

**Strengths:**

- The technical details are clearly stated about the semantic consistency and structure correlation.
- A memory-based learning structure is proposed to facilitate appearance encoding, storage, selecting and consistent learning.
- A main body-aware structure correlation method is proposed to enable recognizing cartoon characters with structural complexity.

**Limitations:**

- The technical contributions of this work are increment. The presented methodology and technology in Representation Consistency are pretty common.
-  In structure correlation, the presented idea is similar to this work CPNet [20].
-  In evaluation, the current results can't fully verify the effectiveness of the proposed CartooNet. To be specific, the performance gain is weak on the dataset CartoonSet, and the results of the CartooNet are inferior to the baselines on the LIP dataset(88.59% vs 89.05%、69.57 vs 71.49 and 58.27 vs 60.30 w.r.t., Pixel Acc、Mean Acc and Mean IoU )

**Suitability:**

3

---

### Meta-Review · Area_Chair_jV8j · 2024-06-30

**Recommendation:** Accept (Poster)
**Confidence:** 5

**Metareview:**

The draft received one weak reject, one borderline reject and one weak accept for initial ratings. Main concerns were raised on technique clarity, relation to previous works, generalization, etc. The authors provided a rebuttal, which did not convince the first reviewer but successfully addressed issues raised in the other two reviewers. As a result, the final ratings were boosted to borderline reject, border accept and accept. Checking all the reviews and authors' feedback, we agree with the majoratiy of the ratings and would recommend the work to ACM MM.